# Relationship between Cardiovascular Risk Factors and Composite Cardiovascular Outcomes in Patients Hospitalized with Takotsubo Syndrome: A Nationwide Analysis

**DOI:** 10.3390/medsci11030062

**Published:** 2023-09-21

**Authors:** Nanush Damarlapally, Rupak Desai, Aanchal Sawhney, Jyoti Verma, Harroop Singh Klair, Dhanush Kolli, Birimroz Singh Sibia, Vardhan Chalasani, Rasya Reddy, Jithin Kolli, Ikechukwu Ogbu, Jyotsna Gummadi

**Affiliations:** 1Department of Health Sciences, Houston Community College (Coleman), Houston, TX 77030, USA; nanushraja@gmail.com; 2Independent Researcher, Atlanta, GA 30033, USA; drrupakdesai@gmail.com; 3Department of Internal Medicine, Crozer Chester Medical Center, Upland, PA 19015, USA; draanchalsawhney@gmail.com; 4Department of Internal Medicine, North Alabama Medical Center, Florence, AL 35630, USA; jyoti.v9426@gmail.com; 5Department of Medicine, Government Medical College, Patiala 147001, India; harroop100@gmail.com; 6Department of Medicine, Kasturba Medical College, Manipal 576104, India; dhanushkolli7515@gmail.com (D.K.); birimrozsinghsibia1996@gmail.com (B.S.S.); vardhan2505@gmail.com (V.C.); rasyareddy143@gmail.com (R.R.); 7Department of Medicine, Jagadguru Sri Shivarathreeshwara Medical College, Mysore 570015, India; kollijithin7515@gmail.com; 8Department of Internal Medicine, Mountainview Hospital Sunrise GME, Las Vegas, NV 89128, USA; iogbu832267@gmail.com; 9Department of Medicine, Medstar Franklin Square Medical Center, Baltimore, MD 21237, USA

**Keywords:** Takotsubo syndrome, cardiovascular disease risk, hypertension, diabetes, hyperlipidemia, smoking, obesity, mortality, cardiogenic shock, stroke, arrhythmia

## Abstract

The association of traditional cardiovascular disease (CVD) risk factors with outcomes of Takotsubo syndrome (TTS) is not well-defined. This study examined how modifiable CVD risk factors affect composite cardiovascular outcomes in TTS hospitalizations. TTS admissions were identified using ICD-10 codes and compared for demographics and comorbidities using the 2019 National Inpatient Sample. A multivariable regression examined the association of traditional CVD risk variables with adverse composite cardiovascular outcomes in TTS, controlling for confounders including sociodemographic or hospital-level characteristics and other relevant comorbidities. A total of 16,055 (38.1%) of the 41,855 adult TTS admissions had composite cardiovascular outcomes (TACCO). The TACCO cohort was 81.5% white, 77.3% female, and 72 years old. This group had higher rates of diabetes and peripheral vascular disease (PVD). The results showed that a higher prevalence of diabetes with chronic complications (OR = 1.18) and complicated hypertension (HTN) (OR = 1.1) predicted TACCO, whereas tobacco use disorder (OR = 0.84), hyperlipidemia (OR = 0.76), and uncomplicated HTN (OR = 0.65) (*p* < 0.001) showed a paradoxical effect with TACCO. TACCO had fewer routine discharges (35.3% vs. 63.4%), longer stays (6 vs. 3 days), and higher median hospital costs (78,309 USD vs. 44,966 USD). This population-based study found that complicated HTN and DM with chronic complications are strongly associated with adverse cardiovascular outcomes in TTS hospitalizations. But still, some risk factors, such as hyperlipidemia and uncomplicated HTN, have counterintuitive effects that require further evaluation. To prevent cardiac events in TTS patients, traditional CVD risk factors must be addressed.

## 1. Introduction

Diseases of the cardiovascular system continue to be one of the most important issues in public health, as they are responsible for one of the major causes of death across the world. Among the various cardiovascular diseases, Takotsubo syndrome (TTS) has emerged as an interesting entity requiring an increasing amount of attention from both clinicians and researchers. Due to the fact that its clinical presentation often resembles acute coronary syndrome, making a correct diagnosis may be an extremely difficult task. Takotsubo syndrome, which is commonly known as stress-induced cardiomyopathy or broken heart syndrome, is characterized by a sudden and temporary weakening of the myocardium. This leads to a form of left ventricular dysfunction, initiated by mental or physical stress, often without any obvious signs of coronary artery disease. Beyond its distinctive cardiac manifestations, patients with TTS frequently share a common thread with individuals at risk for conventional cardiovascular diseases, which include but are not limited to hypertension, diabetes mellitus, and hyperlipidemia. The convergence of these risk factors in TTS patients suggests a multifaceted relationship, hinting at the possibility that the presence and severity of these factors may influence the courses and outcomes of this perplexing syndrome. In the past, TTS was considered a self-limiting condition. However, recent evidence suggests that various factors can contribute to cardiovascular complications, including cardiogenic shock and even mortality [1]. Between 2007 and 2012, there was a roughly threefold increase in the number of people who were hospitalized due to TTS [1,2]. Research that was conducted on a national scale found that the costs of medical treatment associated with TTS amounted to $112 million, while the percentage of patients who required readmission within thirty days was close to 10% [3]. Moreover, during COVID-19, the prevalence of TTS was observed among people with a mean age of 70.8 ± 15.2 years, especially among the elderly and females [4]. The causes could be due to psychosocial stressors such as social isolation, financial strain, and anxiety related to both the viral infection and its novel vaccines [5,6]. Despite the fact that a number of conventional cardiovascular disease risk factors (CVD), such as hypertension (HTN), diabetes (DM), hyperlipidemia, tobacco use disorder, and obesity, have been linked to the development and prognosis of TTS [7], their specific relationship with TTS outcomes still needs further exploration.

In the face of an increasing incidence of TTS and its potential complications, it becomes crucial to identify patients at high risk, examine the prevalence, demographics, and comorbidities associated with TTS, and find discrepancies in the risk factors associated with adverse composite cardiovascular outcomes [8,9]. By delving deeply into this relationship, we aim to cast a spotlight on the underlying pathophysiological mechanisms, prognostic implications, and therapeutic opportunities specific to TTS. This endeavor carries profound significance, not only for enhancing the clinical management and understanding of TTS, but also for the broader field of cardiovascular research. In our study, we aim to augment our collective understanding of this complex condition and its associated factors, thereby informing clinical decision-making and fostering improved patient outcomes by establishing proper management strategies.

## 2. Materials and Methods

### 2.1. Data Extraction and Study Population

We used the National Inpatient Sample (NIS) database from 2019 to conduct this retrospective analysis. The NIS database, the largest all-payer inpatient healthcare database in the United States, includes data on more than seven million hospital stays each year. It covers patient demographics, comorbidities, procedures, outcomes, and hospital characteristics. Our study included adult patients (age ≥ 18 years) who were hospitalized with TTS. These patients were identified using the International Classification of Diseases, Tenth Revision, Clinical Modification (ICD-10-CM) code I51.81 (Figure 1). The ICD-10 CM is a system used by physicians and other healthcare providers to classify and code all diagnoses, symptoms and procedures recorded in association with hospital care in the United States. The codes are developed and maintained by the Center for Disease Control and Prevention’s (CDC) National Center for Health Statistics under authorization by the World Health Organization (WHO). Cardiovascular and extracardiac comorbidities were identified using the Elixhauser Comorbidity Indices and predefined comorbidities in the available database.

### 2.2. Outcomes and Data Synthesis

The primary outcome was the occurrence of composite adverse cardiovascular outcomes (TACCO) during hospitalization, including all-cause mortality, cardiogenic shock, arrhythmia, stroke, and venous thromboembolism (VTE). Covariates included demographic characteristics (age, sex, race), insurance status, hospital characteristics (region, bed size, teaching status), and comorbidities that are known to be associated with TTS or traditional cardiovascular disease (CVD) risk factors. The comorbidities of interest included hypertension (HTN), diabetes mellitus (DM), hyperlipidemia, tobacco use disorder, and obesity.

### 2.3. Statistical Analysis

Descriptive statistics were used to summarize the baseline characteristics of the study population. We used chi-square tests to compare the TTS admissions with and without TACCO for categorical variables and the Mann–Whitney U test for continuous variables [non-normal distribution]. A multivariable logistic regression analysis was performed to identify traditional cardiovascular disease risk factors [hypertension, diabetes, hyperlipidemia, obesity, and tobacco use disorder] independently predicting TACCO, controlling for age, sex, race, payer status, income quartile, type of admission, hospital characteristics, and other relevant comorbidities. We used odds ratios (ORs) with 95% confidence intervals (CIs) to quantify the strength of the association between the covariates and TACCO. All analyses used weighted data and complex survey modules in IBM SPSS Statistics version 25.0. A *p*-value of less than 0.05 was the threshold for statistical significance. The Healthcare Cost and Utilization Project (HCUP) privacy policy prevented publishing cell sizes under 11 [10].

## 3. Results

### 3.1. Study Selection

We identified 41,855 adult TTS admissions from the National Inpatient Sample 2019 database. Among these admissions, 16,055 (38.3%) had composite adverse cardiovascular outcomes (TACCO) during hospitalization. Figure 1 illustrates a comprehensive search and analysis strategy.

### 3.2. Baseline Characteristics

The TACCO cohort was found to be predominately composed of females (77.3% vs. 71.5%), whites (81.5% vs. 73.7%), Medicare payers (67.6% vs. 61%), and those with a Nationwide Median Household Income in the third quartile range (27% vs. 26.6%) with comorbidities such as diabetes mellitus (DM) (46.5% vs. 35.2%), peripheral vascular disease (PVD) (12.3% vs. 6.2%), and chronic kidney disease (CKD) (8.3% vs. 5.3%) compared with the non-TACCO cohort (all *p* < 0.001). Table 1 shows the baseline characteristics of TTS admissions with and without TACCO.

### 3.3. CVD Risk Factors and Their Effect on Outcomes

Table 2 presents the results of the multivariable logistic regression analysis of the association between traditional CVD risk factors and TACCO. After adjustments for psychosocial factors, demographic factors, prior MI, prior PCI, prior coronary artery bypass graft, drug abuse, and various systemic diseases, we observed that the TTS patients having DM with chronic complications (OR = 1.18, 95% CI = 1.12–1.23), and those with complicated hypertension (HTN) (OR = 1.14, 95% CI = 1.08–1.21) had higher odds of TACCO. In contrast, TTS patients with tobacco use disorder (OR = 0.84, 95% CI = 0.78–0.91), hyperlipidemia (OR = 0.76, 95% CI = 0.72–0.81), and uncomplicated HTN (OR = 0.65, 95% CI = 0.61–0.70) had lower odds of TACCO (all *p* < 0.001). Also, we found that TACCO patients were observed to have fewer routine discharges (35.3% vs. 63.4%), longer hospital stays (6 vs. 3 days), and higher median hospital costs (78,309 USD vs. 44,966 USD). This demonstrates that TTS-related composite cardiovascular outcomes increase resource use and healthcare expenses.

We also explored predictors of individual cardiovascular outcomes in TTS, focusing on all modifiable cardiovascular risk factors. During this multivariate regression analysis, the covariates were adjusted for age at admission, sex, race, the patient’s median household income quartile based on ZIP code, admission type, primary expected payer type, hospital location, and other comorbid conditions. With respect to the all-cause mortality outcome, the predominant predictor risk was 44%, significantly higher for diabetes with chronic complications (OR = 1.44, 95% CI 1.12–1.86, *p* = 0.005), but showed a protective effect for hyperlipidemia, reducing the all-cause mortality significantly by 46% (OR = 0.54, 95% CI 0.44–0.67, *p* < 0.001). As for cardiogenic shock as an outcome, diabetes without chronic complications demonstrated a significant 35% reduced risk (OR = 0.65, 95% CI 0.44–0.96, *p* = 0.032), while hyperlipidemia was associated with a significant 28% lower risk (OR = 0.72, 95% CI 0.60–0.86, *p* < 0.001. Complicated hypertension showed a significant 23% higher risk (OR = 1.23, 95% CI 1.08–1.40, *p* = 0.001) and obesity exhibited a significant 21% higher risk (OR = 1.23, 95% CI 1.08–1.40, *p* = 0.001) for developing dysrhythmias. Conversely, hyperlipidemia exhibited a protective effect, reducing the risk of dysrhythmias significantly by 13% (OR = 0.87, 95% CI 0.78–0.97, *p* = 0.012), whereas obesity as a predictor increased the risk of dysrhythmias significantly by 21% (OR = 1.21, 95% CI 1.04–1.42, *p* = 0.016). When stroke was an outcome, complicated hypertension significantly increased the risk by 34% (OR = 1.34, 95% CI 1.04–1.73, *p* = 0.025). Conversely, hyperlipidemia exhibited a protective effect, reducing the stroke risk significantly by 24% (OR = 0.76, 95% CI 0.62–0.93, *p* = 0.007). Lastly, in the context of cardiovascular outcomes for acute VTE, obesity was identified as a significant predictor, increasing the risk by 61% (OR = 1.61, 95% CI 1.16–2.23, *p* = 0.005), while complicated hypertension reduced the acute VTE risk significantly by 27% (OR = 0.73, 95% CI 0.54–0.98, *p* = 0.037), and hyperlipidemia also significantly reduced the acute VTE risk by 30% (OR = 0.7, 95% CI 0.54–0.9, *p* = 0.005). All related information can be found in [Appendix A.

Despite being recognized as traditional CVD risk factors, hyperlipidemia and uncomplicated hypertension demonstrated paradoxical impacts on TTS-associated composite cardiovascular outcomes, which calls for additional research to determine their impact on patient outcomes. The pathophysiology of these disorders may affect TTS progression and TACCO risk.

## 4. Discussion

The National Inpatient Sample 2019 was used to examine how traditional CVD risk factors affect poor composite cardiovascular outcomes in TTS patients. These outcomes included all-cause mortality, cardiogenic shock, arrhythmia, stroke, and venous thromboembolism (VTE). Among the TACCO cohort, we observed a predominance of female patients, an older age profile, and a higher prevalence of comorbidities such as diabetes mellitus, peripheral vascular disease (PVD), and chronic kidney disease (CKD) when compared to the non-TACCO group. Healthcare providers must be vigilant regarding the heightened risk associated with conditions like diabetes with chronic complications and complicated hypertension, enabling effective risk stratification and guiding tailored interventions to enhance patient outcomes. Also, 38.3% of TTS admissions had composite cardiovascular outcomes, including all-cause death, cardiogenic shock, arrhythmia, stroke, and VTE, with a stronger association of HTN and DM in TTS hospitalizations. This observation aligns with prior research emphasizing the significance of HTN and DM in predicting adverse outcomes among TTS patients [3,8,11].

In this analysis, we found that complicated hypertension, diabetes with chronic complications, tobacco use disorder, hyperlipidemia, and uncomplicated hypertension had varying associations with TACCO. The intriguing paradoxical effects observed in TTS patients with hyperlipidemia and uncomplicated hypertension necessitate further research into the underlying pathophysiological mechanisms, emphasizing the need to explore how these factors influence TTS progression and their differential impact on specific patient subgroups. This knowledge supports a more patient-centered approach to care, stressing the importance of personalized risk assessment and management strategies that account for individual TTS patients’ comorbidities and risk profiles. Complicated hypertension and diabetes with chronic complications were positively associated with TACCO, indicating that these risk factors may increase the likelihood of adverse cardiovascular events in TTS patients. In the literature, a systematic review of 1109 TTS patients revealed that 54% of TTS patients carry a diagnosis of HTN [7]. Patients with uncomplicated hypertension have a history of hypertension but no cardiovascular incidents at baseline. A study of 6837 TTS patients revealed that there is no association between uncomplicated HTN and the incidence of TTS [12]. However, in a prospective study of 749 TTS patients, it was found that the incidence of HTN with cardiovascular events in the TTS recurrence group was significantly higher than that in the non-recurrence group (86.7% vs. 68.3%) [13].

Research on the impact of DM on TTS outcomes has produced contradictory findings. TTS patients conventionally have diabetes, with prevalence rates ranging from 12.6% to 22.8% [14,15,16]. DM was identified as a predictor of 90-day TTS readmission in retrospective research [17]. However, some studies have stated that DM is unusual among TTS patients, and it has been hypothesized that neuropathy from DM delays the development of TTS [18,19]. In contradiction to the study that showed obese patients exhibited TTS-related complications, a paradoxical obesity study revealed that normal-weight individuals are more predisposed to TTS complications [20,21]. Thus, although incidents of DM prevalence and its complications leading to TTS have been described, a definitive link between the two disorders has not yet been clearly established.

In our study, tobacco use disorder, hyperlipidemia, and uncomplicated hypertension demonstrated a paradoxical effect, suggesting that they may have a protective or mitigating role in TACCO occurrence. Our study detected a counterintuitive effect of hyperlipidemia on composite cardiovascular outcomes. It still remains unclear if medications taken by TTS patients to control hyperlipidemia had any role in the observed paradoxically lower odds of adverse events. While evaluating the associations with individual cardiovascular outcomes, it is worth noting that diabetes accompanied by chronic complications consistently emerged as a factor contributing to an increased risk of all-cause mortality, cardiogenic shock, and stroke in patients with TTS. Surprisingly, hyperlipidemia displayed an effect across these outcomes, lowering the risk of all-cause mortality, cardiogenic shock, and stroke. On the other hand, complicated hypertension was linked to a risk of arrhythmia, dysrhythmia, and stroke, but did not significantly impact cardiogenic shock. Obesity was identified as a predictor for both arrhythmia/dysrhythmia and acute VTE, increasing the risk in both scenarios. Furthermore, there were no associations found between tobacco use disorder and any of the outcomes examined. These findings underline the importance of managing diabetes with complications and hyperlipidemia in cases of TTS, while also acknowledging the varying impacts of other comorbidities and risk factors on different cardiovascular outcomes. Further research is necessary to gain insights into the relationships between these factors and TTS outcomes.

A detailed investigation of 6837 patients diagnosed with TTS revealed a correlation between lipid levels and the development of TTS after considering other existing comorbid conditions [12,22]. These individuals had lower rates of mortality during hospitalization (1.1% vs. 2.4%, *p* = 0.027), fewer cases of respiratory failure (9.1% vs. 12.1%, *p* = 0.022), and a reduced occurrence of cardiogenic shock (3.4% vs. 5.6%). Moreover, those with higher lipid levels experienced shorter hospital stays (mean ± SD; 3.20 ± 3.27 days) compared to the non-hyperlipidemia TTS group (mean ± SD; 3.57 ± 3.14 days). Additionally, hospital charges were lower for patients with hyperlipidemia compared to those without it (*p* = 0.013) [23]. CA. The increased level of lipoproteins in hyperlipidemia patients has been shown to decrease the risk of sepsis, which is responsible for 21.6% of inpatient TTS mortality [24]. By interacting with the bacterial lipopolysaccharides in sepsis, lipoproteins can decrease the inflammatory immune responses [25]. Also, statins, which are used as first-line drugs for hyperlipidemia management, tend to decrease inflammation by decreasing the expression of interleukin-6, which is accountable for most of the adverse events in TTS patients [23,26]. These findings align with research suggesting that elevated cholesterol levels have effects on short- and long-term prognoses following myocardial infarction, while lower serum cholesterol levels may lead to poorer outcomes within the first year after a heart failure event [27]. However, further research is necessary to comprehend the relationship between hyperlipidemia and TTS development as well as its underlying mechanisms.

The study also revealed a correlation between adverse composite cardiovascular outcomes and increased healthcare costs. TACCO patients exhibited fewer routine discharges, longer hospital stays, and higher median hospital costs, thereby indicating that TTS-related composite cardiovascular outcomes escalate resource usage and healthcare expenditures. Seasonal variations, especially winter periods and other extreme weather conditions such as typhoon landfalls and earthquakes, showed increased hospitalizations with associated comorbidities such as AF, VTE, and MI [28,29]. This may be linked to increased physical, psychological, and emotional stresses due to the activation of sympathetic responses. This highlights the potential cost-saving implications of the early detection and treatment of conventional CVD risk factors in patients with TTS. Prioritizing the detection and control of HTN and DM in TTS patients is especially crucial due to their association with increased morbidity and mortality.

However, certain limitations should be acknowledged. We used ICD-10 discharge numbers to identify our cases, and the lack of information about the patients’ presentations limited our ability to independently confirm their diagnosis. The NIS data also lacks information on key clinical indicators of outcomes such as disease severity, left ventricular ejection fraction, medication data, and patients’ initial functional statuses, which may affect the manner in which patient characteristics are adjusted. It is unable to accurately distinguish comorbidities from hospitalization related problems due to the administrative nature of the database resulting from billing and coding errors, and the non-availability of lab results. In addition, the NIS does not provide information that can be used to predict prehospital events or any long-term outcomes after discharge, thereby limiting our study results. This database did not include outpatient data and the role of medications in some of the observed paradoxical associations. Furthermore, the nature of the database and study design cannot help establish the causative role of these CVD risk factors. Nonetheless, the study contributes valuable insights to the existing body of literature on TTS, demonstrating how conventional CVD risk factors influence TTS-associated composite cardiovascular outcomes. Additionally, the study underscores the increased resource utilization and healthcare costs associated with TTS patients experiencing composite cardiovascular outcomes, highlighting the importance of considering these findings in healthcare resource allocation and policy planning to ensure optimal patient outcomes while effectively managing costs. Future research should investigate the counterintuitive impacts of CVD risk factors and their potential preventative or damaging effects on TTS patients. Further exploration of TTS-associated composite cardiovascular outcomes related to other lifestyle changes, such as smoking cessation and weight loss, is also warranted.

## 5. Conclusions

This population-based study provides critical insights into the impact of traditional cardiovascular disease risk factors on composite cardiovascular outcomes in TTS-related hospitalizations. Our findings underscore the strong associations of hypertension and DM with adverse cardiovascular outcomes, highlighting the need for prompt recognition and management of these conditions in patients with Takotsubo syndrome. Interestingly, the study also identified a paradoxical effect of hyperlipidemia and uncomplicated hypertension, warranting further exploration. Despite some limitations, the study significantly contributes to the existing body of literature and advocates for the enhanced evaluation and management of cardiovascular risk factors in TTS patients to improve health outcomes and mitigate healthcare costs.

## Figures and Tables

**Figure 1 medsci-11-00062-f001:**
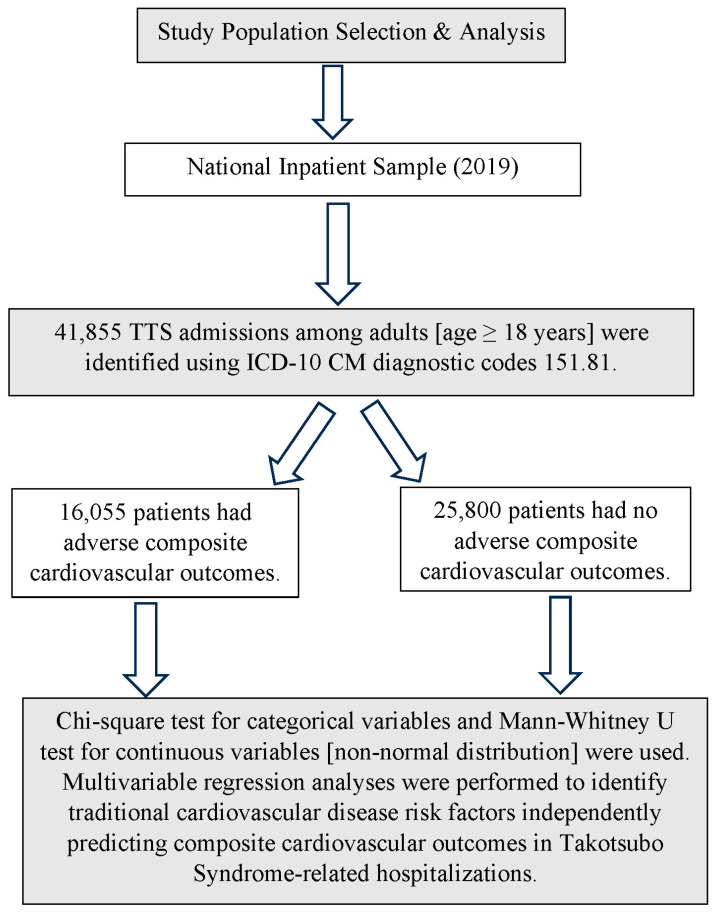
Algorithm of Study Search and Analysis.

**Table 1 medsci-11-00062-t001:** Baseline characteristics of Takotsubo syndrome (TTS) admissions with composite cardiovascular outcomes from 2019.

Variable		TTS with Composite Outcomes		*p*-Value
		No (n = 25,800)	Yes (n = 16,055)	
**Admission Age** **(years)**	Median [IQR]	67 (58–76)	72 (61–80)	<0.001
**Sex n (%)**	Male	3830 (14.8)	3650 (22.7)	<0.001
	Female	21,970 (85.2)	12,405 (77.3)	<0.001
**Race n (%)**	White	20,240 (78.44)	12,705 (79.1)	<0.001
	Black	2125 (8.3)	1170 (7.3)	<0.001
	Hispanic	1670 (6.5)	930 (5.8)	<0.001
	Asian or Pacific Islander	490 (1.9)	335 (2.1)	<0.001
	Native American	130 (0.5)	120 (0.8)	<0.001
**Median Household Income National Quartile for Patient ZIP Code** **n (%)**	0–25th	6560 (25.9)	3840 (24.3)	<0.001
	26–50th	6615 (26.1)	3940 (24.9)	<0.001
	51–75th	6740 (26.6)	4265 (27.0)	<0.001
	76–100th	5445 (21.5)	3775 (23.9)	<0.001
**Primary Expected Payer** **n (%)**	Medicare	15,720 (61.0)	10,850 (67.6)	<0.001
	Medicaid	2850 (11.1)	1415 (8.8)	<0.001
	Private including HMO	5820 (22.6)	3045 (19.0)	<0.001
	Self-Pay	810 (3.1)	355 (2.2)	<0.001
	No Charges	25 (0.1)	325 (2.0)	<0.001
**Comorbidities n (%)**	Hypertension	16,520 (64.0)	9915 (61.8)	<0.001
	Diabetes	5730 (22.2)	3800 (23.7)	0.001
	Hyperlipidemia	12,540 (48.6)	6965 (43.4)	<0.001
	Obesity	3015 (11.7)	1870 (11.6)	0.905
	Peripheral vascular disease	2230 (8.6)	1935 (12.1)	<0.001
	Prior myocardial infarction	3060 (11.9)	1670 (10.4)	<0.001
	Prior PCI	95 (0.4)	65 (0.4)	0.555
	Prior CABG	1155 (4.5)	785 (4.9)	0.051
	Tobacco use disorder	5115 (19.8)	2453 (15.2)	<0.001
	Drug Abuse	1515 (5.9)	785 (4.9)	<0.001
	Alcohol Abuse	1560 (6.0)	860 (5.4)	0.003
**Disposition of patient** **n (%)**	Routine	16,370 (63.4)	5665 (35.3)	<0.001
	Transfer to short-term hospital	580 (2.2)	490 (3.1)	<0.001
	Other transfers including SNF, ICF	4525 (17.5)	4280 (26.7)	<0.001
	Home healthcare	4005 (15.5)	2750 (17.1)	<0.001
**Length of Hospital stay (days)**	Median [IQR]	3 (2–6)	6 (3–11)	<0.001
**Total charges (USD)**	Median [IQR]	44,966 (28,306–80,076)	78,309 (41,253–165,892)	<0.001

n: Number of subjects. HMO: Health Maintenance Organization. PCI: Percutaneous Coronary Intervention. CABG: Coronary Artery Bypass Grafting. SNF: Skilled Nursing Facility. ICF: Intermediate Care Facility.

**Table 2 medsci-11-00062-t002:** Multivariable regression to assess the modification effect of comorbidities on composite adverse cardiovascular outcomes in TTS admissions (all-cause mortality, cardiogenic shock, arrhythmia, stroke, and VTE).

Risk Factor		Odds Ratio	95 Confidence Interval		*p*-Value
			Lower	Upper	
Hypertension, complicated	Yes vs. No	1.14	1.02	1.29	0.027
Hypertension, uncomplicated	Yes vs. No	0.65	0.58	0.73	<0.001
Diabetes with complications	Yes vs. No	1.18	1.02	1.36	0.027
Diabetes *w*/*o* complications	Yes vs. No	0.87	0.73	1.04	0.117
Hyperlipidemia	Yes vs. No	0.76	0.68	0.83	<0.001
Obesity	Yes vs. No	1.14	0.98	1.32	0.089
Tobacco Use Disorder	Yes vs. No	0.84	0.74	0.96	0.012

Dependent Composite TTS Outcomes, *p* < 0.05 indicates statistical significance. MV regression analysis was adjusted for age, sex, race, payer status, income quartile, type of admission, hospital characteristics, alcohol abuse, arthropathies, depression, PVD, prior MI, prior PCI, prior CABG, drug abuse, COPD, hypothyroidism and other thyroid disorders, anxiety and fear-related disorders, cancer, and traditional CVD risk factors. TTS: Takotsubo syndrome. VTE: Venous Thromboembolism. MV: Multivariable. PVD: Peripheral Vascular Disease. MI: Myocardial Infarction. PCI: Percutaneous Coronary Intervention. CABG: Coronary Artery Bypass Graft. COPD: Chronic Obstructive Pulmonary Disease. CVD: Cardiovascular Vascular Disease. *w*/*o*: without.

## Data Availability

The data utilized in this research are available from the author upon request. The information is not available to the public because of privacy restrictions.

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
