# Peer review of "Relationship between Cardiovascular Risk Factors and Composite Cardiovascular Outcomes in Patients Hospitalized with Takotsubo Syndrome: A Nationwide Analysis"

_medsci, 2023, doi:10.3390/medsci11030062_

Round 1
Reviewer 1 Report
The Authors reported a strong association of DM and hypertension with adverse cardiovascular outcome; on top of that great emphasis has been placed on a paradoxical impact of hyperlipidemia and uncomplicated hypertension. In my opinion a significant limitation of this study, already signaled by the Authors, is the lack oflong-term outcomes data. Other observations and suggestions:
- The Authors used a composite outcome, which included all-cause mortality, cardiogenic shock, arrhythmia, stroke, and venous thromboembolism (VTE). It could be interesting to show the association with each single component.
- In table 1 the graphic association of p-value is slightly unclear, sometimes every variable has a value, sometimes the value is in the middle.
- I found that the discussion is somewhat unbalanced and chaotic; this part is too long in comparison with the rest of the paper, and focused more on describing other studies, jumping from on one another, overshadowing this study itself.
Author Response
Dear Reviewer,
please see the attachment for response. Thank you for your valuable time in reviewing our manuscript.
Thanks,
Jyotsna Gummadi.

Reviewer 2 Report
I read with great attention the article entitled:
Relationship Between Cardiovascular Risk Factors and Compo-2 site Cardiovascular Outcomes in Patients Hospitalized with Takotsubo Syndrome: A Nationwide Analysis by Nanush Damarlapally et al.
The article is interesting and includes data from a retrospective database. The results are sound and reported the association with the Takotsubo syndrome. The authors should also consider that not all Takotsubo were recognized in the limitations.
My primary concern is about the IRB not being reported, and that precludes publication at the moment.
Best Regard
minor changes
Author Response
Dear Reviewer,
Please see the attachment below. Thank you for your time in reviewing our manuscript.
Thanks,
Jyotsna Gummadi.
